# ROYAL SOCIETY
# OPEN SCIENCE

environmental chemistry/materials science/ environmental science

molecularly imprinted polymer, β-cyclodextrin, adsorption, bisphenol A, water samples

**Author for correspondence:**
N. N. M. Zain
e-mail: nurnadhirah@usm.my

This article has been edited by the Royal Society of Chemistry, including the commissioning, peer review process and editorial aspects up to the point of acceptance.

# Removal of bisphenol A from aqueous media using a highly selective adsorbent of hybridization cyclodextrin with magnetic molecularly imprinted polymer

S. Mamman[1,4], F. B. M. Suah[1], M. Raaov[3], F. S. Mehamod[5], S. Asman[6] and N. N. M. Zain[2]

[1]School of Chemical Sciences, Universiti Sains Malaysia, 11800 Penang, Malaysia
[2]Integrative Medicine Cluster, Advanced Medical and Dental Institute, Universiti Sains Malaysia, 13200 Penang, Malaysia
[3]Department of Chemistry, Faculty of Science, Universiti Malaya, 50603 Kuala Lumpur, Malaysia
[4]Faculty of Natural and Applied Sciences Department of Chemistry, Nasarawa State University Keffi, PMB 1022 Keffi, Nasarawa, Nigeria
[5]Advanced Nano Materials (ANoMA) Research Group, School of Fundamental Science, Universiti Malaysia Terengganu, 21030 Kuala Nerus, Terengganu, Malaysia
[6]Department of Physics and Chemistry, Faculty of Applied Sciences and Technology, Universiti Tun Hussein Onn Malaysia, UTHM Pagoh Campus, Pagoh Higher Education Hub, 84600 Muar, Johor, Malaysia

In this study, a unique magnetic molecularly imprinted polymer (MMIP) adsorbent towards bisphenol A (BPA) as a template molecule was developed by bulk polymerization using β-cyclodextrin (β-CD) as a co-monomer with methacrylic acid (MAA) to form MMIP MAA–βCD as a new adsorbent. β-CD was hybridized with MAA to obtain water-compactible imprinting sites for the effective removal of BPA from aqueous samples. Benzoyl peroxide and trimethylolpropane trimethacrylate were used as the initiator and cross-linker, respectively. The adsorbents were characterized by Fourier transform infrared spectroscopy, scanning electronic microscopy, transmission electron microscopy, vibrating sample magnetometer, Brunauer–Emmett–Teller and X-ray diffraction. [1]H nuclear magnetic resonance spectroscopy was used to characterize the MAA–βCD and BPA–MAA–βCD complex. Several parameters influencing the adsorption efficiency of BPA such as adsorbent dosage, pH of sample solution, contact time, initial concentrations and temperature as well as selectivity and reusability study have been evaluated. MMIP MAA–βCD

showed significantly higher removal efficiency and selective binding capacity towards BPA compared to MMIP MAA owing to its unique morphology with the presence of β-CD. The kinetics data can be well described by the pseudo second-order kinetic and Freundlich isotherm and Halsey models best fitted the isotherm data. The thermodynamic studies indicated that the adsorption reaction was a spontaneous and exothermic process. Therefore, MMIP based on the hybrid monomer of MAA–βCD shows good potential of a new monomer in molecularly imprinted polymer preparation and can be used as an effective adsorbent for the removal of BPA from aqueous solutions.

## 1. Introduction

The release into water sources of a wide variety of chemical pollutants is an ecological problem because of the conceivable toxicity to humans. Because the demand for polycarbonates and epoxy resins is increasing, bisphenol A (BPA) is one of the major chemical products manufactured worldwide, as the market for polycarbonates and epoxy resins is growing [1]. From 1970 to 2013, plastics production increased from 50 million to 300 million tons. The polymer industry experienced a tremendous increase in global revenues from £126 billion in 1970 to over £2.9 trillion [2]. Waste water treatment plants leak BPA into effluent accidentally as these tenacious and xenobiotic compounds were not originally intended to be handled [3]. Owing to the endocrine disruption property of BPA, it can imitate the body's hormones, and normal cell function can be disrupted and can affect human health [4]. BPA can be found in many daily used items including plastic bottles, metal food containers, detergents, fire retardants, food, toys, beauty care products and pesticides. They can also be detected in water sources [5]. The maximum acceptable dose for BPA was established by the US Environmental Protection Agency (USEPA) at $50\,\mu g\,kg^{-1}$ body weight day$^{-1}$, and the tolerable daily intake of $50\,\mu g\,kg^{-1}$ body weight day$^{-1}$ was established as safe for consumption by the European Food Safety Authority (EFSA) [6]. The Health Canada's estimated tolerable daily intake for BPA is $25\,\mu g\,kg^{-1}$ body weight day$^{-1}$ [4,7]. The development of new technologies that are both efficient and selective for the removal of pollutants is of great concern in this regard.

New innovations are consistently being sought for the removal of hazardous pollutants in aqueous samples. Existing methods available for the treatment of pollutants are distillation, coagulation, electrocoagulation [8], biological treatment, catalytic oxidation, ozonation [9], solvent extraction and adsorption [10]. However, some researchers found the method of adsorption superior to other techniques because it offers effective and rapid contaminant removal, along with low operating costs, simple design, easy to use, and less harmful by-products are generated [11]. Physical adsorption is commonly known as a proficient mechanism for removing such types of organic molecules in aqueous media, provided that adsorbents provide large internal or external surfaces that are accessible [3]. Nevertheless, it has been generally agreed that the key factor for effective adsorption is the choice of adsorbent [12].

Recently, materials based on magnetic iron oxides have become a promising alternative adsorbent because the technique may enhance the adsorption process by eliminating the filtration and centrifuging steps in other methods [6,7] and also provides a wide surface area, separation properties and low toxicity [13]. Nanoparticles of iron oxides have been widely used to eliminate BPA and to further isolate adsorbents from matrix samples [14–18]. While the wide area-to-volume ratio of nanomaterials brings about an exponential increment in adsorption capacity, the incidence of aggregation, non-specificity and low stability can restrict the usage of such nanotechnologies owing to the absence of functionality [16]. Molecularly imprinted polymers (MIPs) have gained considerable attention as tailor-made adsorbents for specific recognition of a target molecule and its structurally related compound [17]. It is obtained through the polymerization of functional and cross-linking monomers around the target molecule (template) in a complementary fashion, bringing about an exceptionally three-dimensional network [19,20]. When the template is removed, it creates recognition binding sites in the polymer matrix with specific affinity to the template/target analyte of interest; hence, the template can be recognized from complex environmental samples [20]. Regarding the physical properties and appearance of MIPs, they can be generated from a broad selection of protocols, which allows these materials to be obtained in a wide variety of formats, from macro–micro–nano sizes and thick to thin subnanometer layers [21]. The most widely used technique for preparing MIPs is non-covalent imprinting owing to its relative simplicity on the experimental level [22]. In this process, the complex of the template and functional monomer is formed *in situ* by non-

covalent interactions, such as hydrogen bonding, electrostatic forces, Van der Walls forces and hydrophobic interactions [23]. Many researchers have investigated MIP for BPA [24–29]. Most of these studies used conventional functional monomers such as methacrylic acid (MAA), acrylamide and pyridine 4-vinyl. As a new generation of functional monomers, β-cyclodextrin (β-CD) hybrid monomers have some unique characteristics compared to traditional monomers, such as the creation of inclusion complexes by host–guest interaction [30]. Second, the β-CD unit can form a complex with the target analyte through various types of intermolecular interactions such as Van der Waals force, hydrophobic interaction, electrostatic affinity, dipole–dipole interaction and hydrogen bond interaction during the imprinting phase because of the rigidity and chirality of its hydrophobic cavity. It is a promising step to modify the functional monomer with β-CD, as the cavities are formed by imprinting lack unique binding sites. The recognition capability of the β-CD could be enhanced by linking multiple functional groups of the monomer to β-CD, which improves its binding capacity [31].

In this study, a profoundly specific and proficient magnetic MIP (MMIP) dependent on MAA–βCD hybridization was developed via the bulk polymerization approach. The involvement of magnetic components in the adsorbents created a controllable rebinding process which enhanced the convenient and economical replacement of centrifugation and filtration steps by magnetic separation. To the best of our knowledge, MMIP MAA–βCD for the removal of BPA has not been reported in the literature.

# 2. Experimental

## 2.1. Materials and reagents

Bisphenol A (99%), trimethylolpropane trimethacrylate (TRIM), β-CD, MAA, toluene 2,4-diisocyanate (TDI), dimethylacetamide (DMAC), 2,4-dichlorophenol (2,4-DCP), 2,4-dinitrophenol (2,4-DNP) and dibutyltin dilaurate (DBTDL) were bought from Sigma Aldrich USA. Benzoyl peroxide (BPO), methanol (MeOH), iron (II) chloride tetrahydrate ($FeCl_2.4H_2O$) and iron (III) chloride hexahydrate ($FeCl_3.6H_2O$) were bought from R&M (Essex, UK). Analytical grade absolute ethanol (denatured, 99.7) and methanol were obtained from HmbG Chemicals (Cologne, Germany), and $NH_3$ solution (28%) and acetic acid were purchased from Bright Chem SDN. BHD. Nitrogen gas was obtained from Malaysian oxygen (MOx). The stock solution of 1000 mg l$^{-1}$ BPA, 2,4-DCP and 2,4-DNP was prepared by diluting respective standards in methanol and stored at 4°C. Fresh working standards were prepared daily by diluting the stock solution in deionized water. All reagents and chemicals were of analytical reagent grade and were used as received without further purification.

## 2.2. Characterization techniques

A 2000 Fourier transform infrared (FTIR) spectrometer (Perkin Elmer) was used to record the FTIR spectra of all-synthesized adsorbent materials. Before investigation, the adsorbents were blended in with KBr powder and compressed into pellets. The analysis for liquid samples was carried out using a Bruker FTIR fitted with an attenuated total reflectance (ATR) diamond (Billerica, USA). The morphology of the materials was evaluated by scanning electronic microscope (SEM) Quanta FEG 650 model. Images from the transmission electron microscopy (TEM) were captured to study the structure and the pore size distribution of the materials. A modest quantity of the adsorbent material was dispersed in a few millilitres of methanol in an ultrasonic bath and sonicated for 30 min, and a drop of this dispersed sample was placed on a copper grid coated with a holey carbon film. Before examining with the Philips CM-12 TEM instrument, the samples were dried at room temperature. The sample was scanned at all zones before the picture was taken. X-ray diffraction (XRD) analysis was performed on a Panalytical empyrean model. All the samples were analysed under Cu Kα radiation ($\lambda = 1.5418 \ \Delta$). The full width at half maximum was obtained from the most intense peak using the origin software. Brunauer–Emmett–Teller (BET) analysis was carried out to determine the surface area and the pore diameter of the adsorbent materials. First, the materials were degassed to remove moisture at 80°C before further analysis. A vibrating sample magnetometer (VSM) using lakeshore/ 7404 (lakeshore cryotronics) was used to determine the magnetic strength of the materials. To obtain the proton nuclear magnetic resonance (NMR) spectra of both monomer and complex, about 1 mg of each was dissolved in 1 ml of deuterated dimethyl sulfoxide (DMSO-d6) in a vial and then transferred into the NMR tubes before analysing with a Bruker-Avance 500 MHz spectrometer.

## 2.3. Synthesis of methacrylic acid-β-cyclodextrin monomer

MAA–βCD monomer was prepared depending on the stoichiometric ratios of 0.5 M MAA:1 M TDI:0.5 M β-CD. This proportion was similar to that of the previously published study with slight modification [31,32]. First, MAA and TDI were mixed in 40 ml dimethylacetamide (DMAC) solvent; then, 0.1% dibutyltin dilaurate (catalyst) was applied, and the solution was stirred magnetically in an inert atmosphere for 1 h at room temperature. Then, an amount of β-CD was added along with 10 ml DMAC to this solution mixture and further stirred for 2 h.

## 2.4. Synthesis of magnetic molecularly and non-molecularly imprinted polymers

MMIP MAA–βCD was prepared by the bulk polymerization method according to the literature with minor modifications [32]. First, the template molecule, BPA (0.14 mmol) was dissolved in 20 ml DMAC in a flask. Then, the functional monomer, MAA–βCD or MAA (0.56 mmol), the cross-linker, TRIM (2.80 mmol) and the initiator BPO (1 g) were added into the flask. The contents of the flask were purged for 10 min with nitrogen gas and then sealed and allowed to polymerize for 24 h in a water bath at 70°C. The product was crushed, grounded and sieved to obtain regular-sized polymer particles.

Furthermore, the MMIP/magnetic non-molecularly imprinted polymer (MNIP) was synthesized via a one-step co-precipitation method under a nitrogen atmosphere. The ratios of MIP/NIP : $FeCl_2.4H_2O:2FeCl_3.6H_2O$ were weighed and mixed in 100 ml of deionized water and then stirred for 30 min at 1200 r.p.m. A 12 ml of $NH_3$ solution was added, and then, the temperature was increased to 90°C and the reaction mixture was stirred for 1 h. The resulting magnetic particles were washed with ethanol and ultrapure water to remove unreacted particles. The product was isolated with the application of an external magnet and dried in an oven at 60°C.

Then, the template was removed by washing the polymer with the methanol/acetic acid (9 : 1 v/v) solution. The MMIP was washed several times until no BPA was detected at 276 nm using a UV-visible (UV-vis) spectrophotometer. The particles were then washed several times with ultrapure water. For the preparation of MNIP, a similar procedure used for MMIP MAA–βCD/MMIP MAA was adopted, except that the template was omitted. The synthesis route of the MMIP MAA–βCD is illustrated in figure 1.

## 2.5. Preparation of methacrylic acid-β-cyclodextrin-bisphenol A complex

The BPA inclusion complex was produced using the conventional kneading method. An equimolar amount of MAA–βCD and BPA was kneaded to form a homogeneous paste for about 30 min in a mortar and pestle with a minimum amount of absolute ethanol (denatured, 99.7%). The product was then dried to a constant weight, and a white complex powder was obtained and later characterized by $^1H$ NMR spectroscopy.

## 2.6. Batch adsorption experiment

For this experiment, 20 mg of the adsorbent was weighed, 10 ml (10 mg $l^{-1}$) of analyte solution was added into a glass vial and sealed tightly. Then, it was placed in a shaker and agitated at 250 r.p.m. and 298 K. The effect of different variables such as adsorbent dosage, pH sample solution, contact time, and initial concentration and temperature were studied. After the adsorption, the supernatant is isolated from the adsorbent using an external magnet. The residual concentration of the supernatant was analysed using a UV–vis spectrometer at 276 nm. Figure 2 shows the removal of BPA via the batch adsorption process. The percentage of removal of BPA was calculated using equation (2.1):

$$\text{removal efficiency} (\%) = \frac{(C_0 - C_f)}{C_0} \times 100, \tag{2.1}$$

where $C_0$ (mg $l^{-1}$) is the initial concentration and $C_f$ is the final concentration (mg $l^{-1}$).

The uptake of BPA at equilibrium is defined as the adsorption capacity $q_e$ (mg g$^{-1}$), and $q_e$ is calculated using equation (2.2):

$$q_e = \frac{[(C_0 - C_f) \times V]}{w}, \tag{2.2}$$

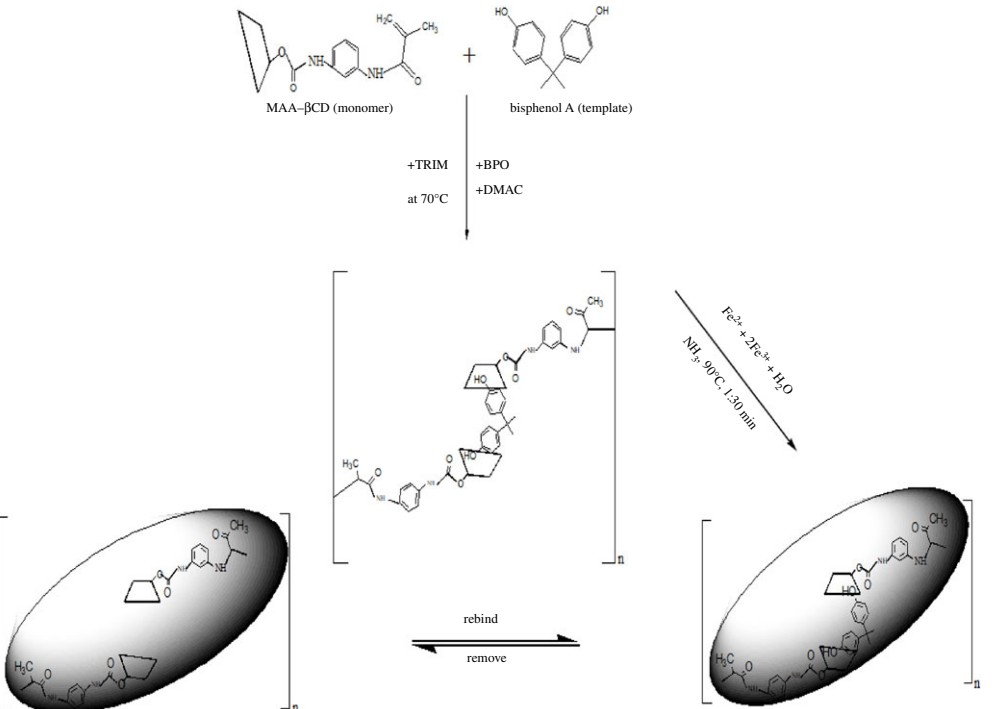

**Figure 1.** Synthesis route of MMIP MAA–βCD.

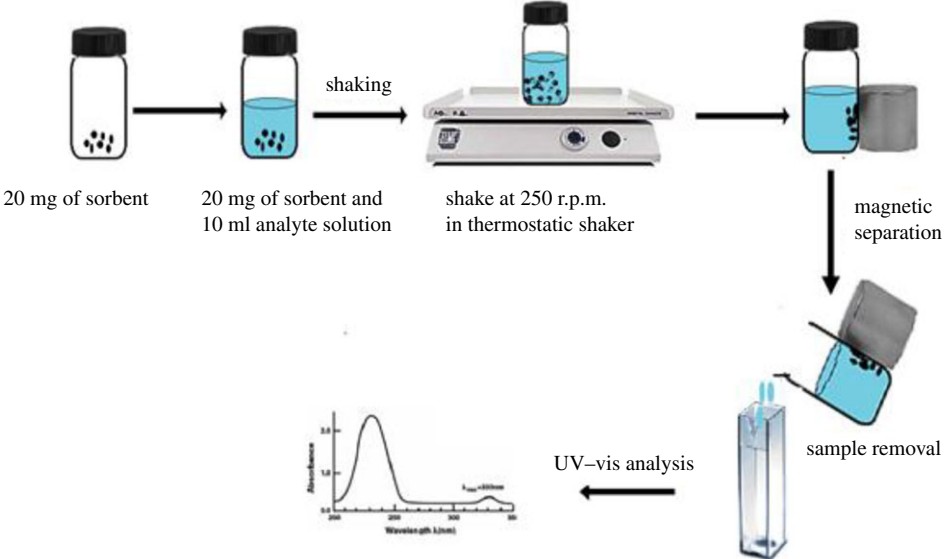

**Figure 2.** Removal of phenolic compounds via the batch adsorption process.

where $C_0$ (mg l$^{-1}$) is the initial concentration, $C_f$ is the final concentration (mg l$^{-1}$). $q_e$ is adsorption capacity (mg g$^{-1}$), $V$ (l) is the volume of the analyte and $w$ (g) is the mass of the adsorbent used.

To evaluate the applicability of the proposed method based on MMIP MAA–βCD using UV–vis spectrometer, real water samples contaminated with BPA were analysed. For the assessment of the accuracy and the precision, real water samples spiked with BPA were tested by the method built.

## 2.7. Selectivity study

Selectivity tests were performed to test the binding potential of both MMIP and MNIP polymers towards the BPA molecule. A 20 mg of MMIP or MNIP is combined with 10 ml of 10 mg l$^{-1}$ of BPA, 2,4-DCP or 2,4-DNP solution (each in different vials) and shaken at 298 K for 60 min. The concentration of phenolic

compounds remaining after adsorption was measured using UV–vis at their respective wavelengths of 276 nm (BPA), 358 nm (2,4-DNP) and 286 nm (2,4-DCP). The studies were conducted in triplicate. The amount of analyte bound to the polymers was determined by subtracting the amount of free analyte from the original amount of the analyte. The values of $K_d$, $K$ and $K'$ are determined using equations (2.3)–(2.5):

$$K_d = \frac{[C_i - C_f]}{m} \times V, \tag{2.3}$$

$$K = K_{d(BPA)}/K_{d(phenolic)}, \tag{2.4}$$

$$K' = K_{MMIP}/K_{MNIP}, \tag{2.5}$$

where $C_i$ (mg l$^{-1}$) and $C_f$ (mg l$^{-1}$) are the initial and final concentrations. $V$ (l) is the volume used, and $m$ (g) is the weight of adsorbents. $K_{d(BPA)}$ and $K_{d(phenolic)}$ are the static distribution coefficients of BPA and competing compounds (2,4-DCP and 2,4-DNP representing phenolic compounds), respectively. $K$ is the selectivity coefficient and $K'$ is the relative selectivity coefficient, while $K_{MMIP}$ and $K_{MNIP}$ are the selectivity coefficients of MMIP and MNIP, respectively.

## 2.8. Reusability study

This experiment was conducted under optimum adsorbent conditions using the same adsorbent (20 mg) for the removal process for six successive cycles. For each application, the adsorbents were washed with methanol, followed by ultra-pure water washing to make the adsorption sites available for the next phase. Then, the adsorbent was dried at 60°C before being reused in the next removal cycle.

## 2.9. Method validation

To verify the applicability of the method, it was evaluated under optimized adsorbent conditions in terms of linearity, precision, and removal percentage.

### 2.9.1. Linearity

The removal procedure was constructed under the optimal conditions of all parameters analysed in the calibration curve. The linear range of the phenolic compounds was developed using serial dilutions at various concentrations.

### 2.9.2. Precision and reproducibility

The reproducibility and precision of the proposed method was presented as inter-day and intra-day analyses, low, medium and high concentrations (1, 10 and 60 mg l$^{-1}$, respectively) levels were used. For the intra-day using triplicate determination for each vial, each concentration level was conducted in five separate vials ($n = 5$) within the same day. Thus, similar procedures were followed for inter-day for five successive days ($n = 6$) to obtain the percentage removal and relative standard deviation (RSD) (%) values.

### 2.9.3. Removal

The samples were spiked at three different concentrations (1, 10 and 60 mg l$^{-1}$) and have been applied to the removal study of the proposed method to calculate its accuracy. The removal was carried out in triplicate ($n = 3$) for each of the concentrations tested. The percentage of removal was computed using equation (2.1).

# 3. Collection and preparation of real samples

Four water samples were obtained from various sites, including tap water from the Advanced Medical and Dental Institute's (AMDIs) integrative laboratory, Universiti Sains Malaysia, and environmental water samples were collected from the rubber, plastic and wood industries from the Prai industrial region of Penang, Malaysia. The samples were filtered through a 0.45 μm cellulose membrane filter and placed at 4°C in the refrigerator until analysis. Under optimal adsorbate conditions, the analytes were then spiked at various concentrations. For accuracy and precision, the batch adsorptions were performed in triplicate ($n = 3$).

# 4. Results and discussion

## 4.1. Characterization of the synthesized material

### 4.1.1. Fourier transform infrared analysis

Figure 3 displays the FTIR spectra of MAA, β-CD, TDI, MAA-TDI and MAA–βCD. The complete disappearance of TDI's N=C=O peak at 2271 cm$^{-1}$ after reaction with β-CD and the emergence of a new carbamate group (–NHCO) peak at 3368 cm$^{-1}$ (e) show that TDI's N=C=O group reacted with β-CD's one of the OH. C=C was allocated 1618 cm$^{-1}$ The presence of this peak indicates that MAA's double bond is still intact; this is important, as it will be used for cross-linking with TRIM during the polymerization process. The result shows that MAA has been successfully hybridized with β-CD.

The FTIR spectra of (a) BPA, (b) MMIP MAA–βCD, (c) MNIP MAA–βCD, (d) MMIP MAA and (e) MNIP MAA are presented in figure 4. The MMIPs/MNIPs have the same characteristic peaks except for slight spectra shifts, which indicate that there was complete template removal and the appearance of the peak at 582 indicated the presence of Fe–O from magnetite. The peaks observed were around 3438 cm$^{-1}$, 3128 cm$^{-1}$, 2938 cm$^{-1}$, 2345 cm$^{-1}$, 1723 cm$^{-1}$, 1629 cm$^{-1}$, 1393 cm$^{-1}$, 1273 cm$^{-1}$ and 1136 cm$^{-1}$. The peak at 3431 cm$^{-1}$ was assigned to O–H stretch and 3128 cm$^{-1}$ corresponds to the –NH group, and the band at 2924 cm$^{-1}$ was owing to the symmetric and asymmetric C–H stretching vibrations. The band at 1723 cm$^{-1}$ indicated that the TRIM cross-linker reaction was successful, and the peak around 1629 cm$^{-1}$ was assigned to C=O stretch, the peak at 1393 cm$^{-1}$ corresponds to O–H bending, the peak at 1273 cm$^{-1}$ was assigned to the stretching vibrations of C–OH of carboxylic acid, the peak at 1136 cm$^{-1}$ and 1405 cm$^{-1}$ was assigned to C–O and the peak at 582 cm$^{-1}$ was owing to Fe–O stretching vibrations.

### 4.1.2. Scanning electronic microscopy analysis

MMIP/MNIP morphology was assessed by SEM. The micrographs acquired are shown in figure 5. The morphology of the obtained SEM micrographs had substantial dissimilarities. From the images, the surface images were extremely different, which offered strong evidence of the successful modification of the polymers in the presence and absence of BPA template. It can be found that the MMIP surface was more porous and rougher than the MNIP surface that could be owing to the presence of imprinted sites [23]. The polymers' shape and morphology greatly influenced their adsorption efficiency, which explains why the MMIP MAA–βCD has a higher percentage of removal than MMIP MAA.

### 4.1.3. Transmission electron microscopy analysis

TEM images were captured at 100 nm to investigate the morphology and size of the materials. The result is depicted in (figure 6). The morphology of the nanoparticles was monodispersed; most of them are quasi-spherical in shape with a smooth surface and uniform size. Some smaller particles aggregate into bigger particles owing to their extremely small sizes.

### 4.1.4. Brunaur-Emmett-Teller analysis

BET was employed to gain further knowledge about the surface areas, pore volumes and mean pore diameter of the synthesized materials using BET and Barrett-Joyner-Halenda (BJH) analyses. BET results of synthesized adsorbent materials are listed in the electronic supplementary material, table SD1. The pore size distribution of the materials was MMIP MAA–βCD (15.24 nm), MNIP MAA–βCD (14.54 nm), MMIP MAA (10.86 nm) and MNIP MAA (10.26 nm), which falls within the mesoporous range with ($d_P$ = 2–50 nm) according to the International Union of Pure and Applied Chemistry classification [33]. MMIP MAA–βCD was found to be 92.14 m$^2$ g$^{-1}$ and thus it is higher when compared to its corresponding MNIP MAA–βCD (79.31 m$^2$ g$^{-1}$) and the unmodified materials such as MNIP MAA and MMIP MAA (73.06 m$^2$ g$^{-1}$ and 77.31 m$^2$ g$^{-1}$, respectively). The nitrogen adsorption/desorption isotherms are shown in the electronic supplementary material, figure SD1. All materials were reported to follow type IV isotherm and type H3 hysteresis loop, confirming the mesoporosity of the polymers with the presence of heterogeneous open pores [34]. This finding essentially implies that the MMIP has a greater surface area owing to the existence of imprinted cavities formed by the template, which left rough surface on the MMIP particles with a higher surface area upon removal.

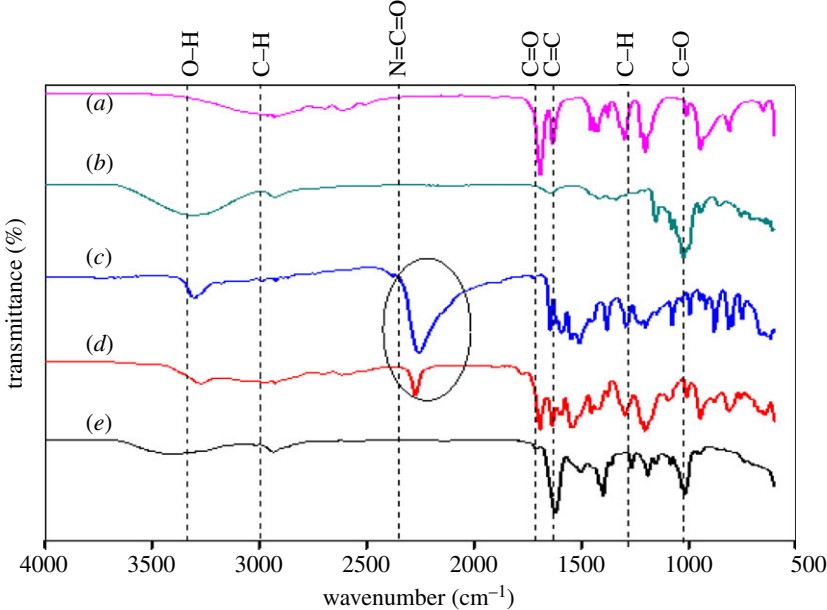

**Figure 3.** FTIR spectra of (*a*) methacrylic acid (MAA), (*b*) β-cyclodextrin (β-CD), (*c*) toluene 2, 4-diisocyanate (TDI), (*d*) TDI-MAA and (*e*) MAA–βCD.

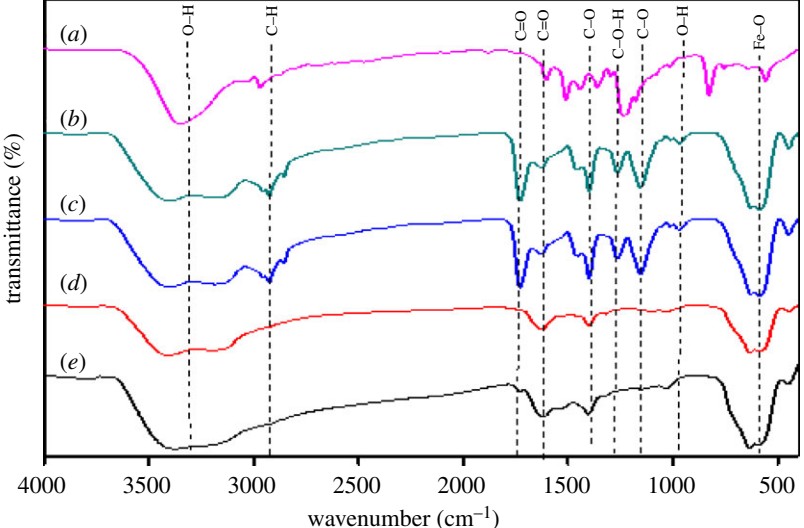

**Figure 4.** FTIR spectra of (*a*) BPA, (*b*) MMIP MAA–βCD, (*c*) MNIP MAA–βCD, (*d*) MMIP MAA and (*e*) MNIP MAA.

### 4.1.5. Vibrating sample magnetometer analysis

The VSM analysis was carried out to investigate the magnetization strength of the synthesized adsorbents. When preparing materials for future magnetic separation, the sample must have adequate magnetic properties for practical use. The magnetic hysteresis curve is shown in figure 7. Magnetic saturation (Ms) values were (*a*) bare $Fe_3O_4$ (80 emu g$^{-1}$), (*b*) MMIP MAA (58 emu g$^{-1}$) and (*c*) MMIP MAA–βCD (24 emu g$^{-1}$). The magnetization obtained in the same field for MMIP MAA–βCD was greater owing to the inclusion of non-magnetic materials (β-CD layer) on the surface of the polymeric magnetic particles. Remarkably, the curve indicated that the materials were supra magnetic and can react efficiently to the external magnetic field, after which it was well dispersed when the magnetic field was removed, as shown in figure 7. The black particles were drawn to the wall of the vial in a short period (about 1 min) after the magnetic field had been applied. This shows, thus, that Ms is adequately sufficient for practical applications for the removal of BPA.

### 4.1.6. X-ray diffraction analysis

This method is used to evaluate the crystalline structure of (*a*) MMIP MAA–βCD, (*b*) MMIP MAA and (*c*) $Fe_3O_4$. The findings of the phase study of $Fe_3O_4$ particles display peaks corresponding to the joint

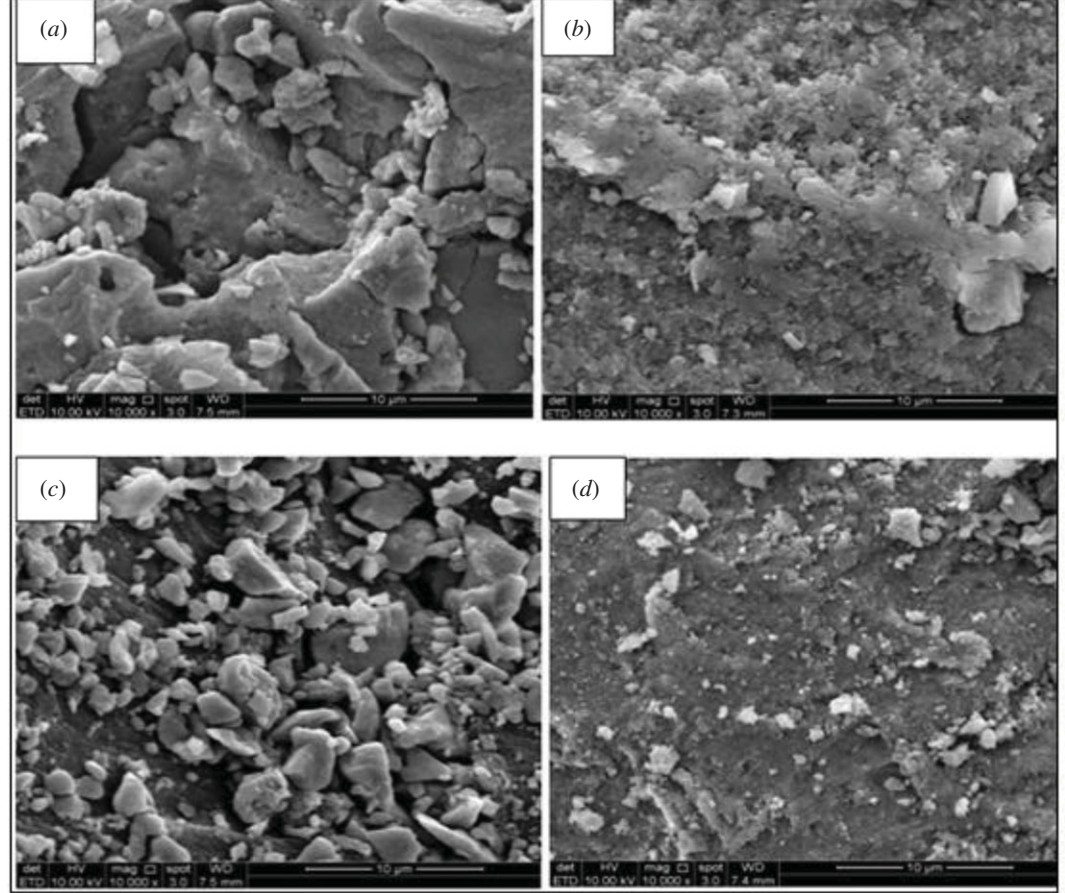

**Figure 5.** SEM micrographs of (*a*) MMIP MAA–βCD, (*b*) MNIP MAA–βCD, (*c*) MMIP MAA and (*d*) MNIP MAA.

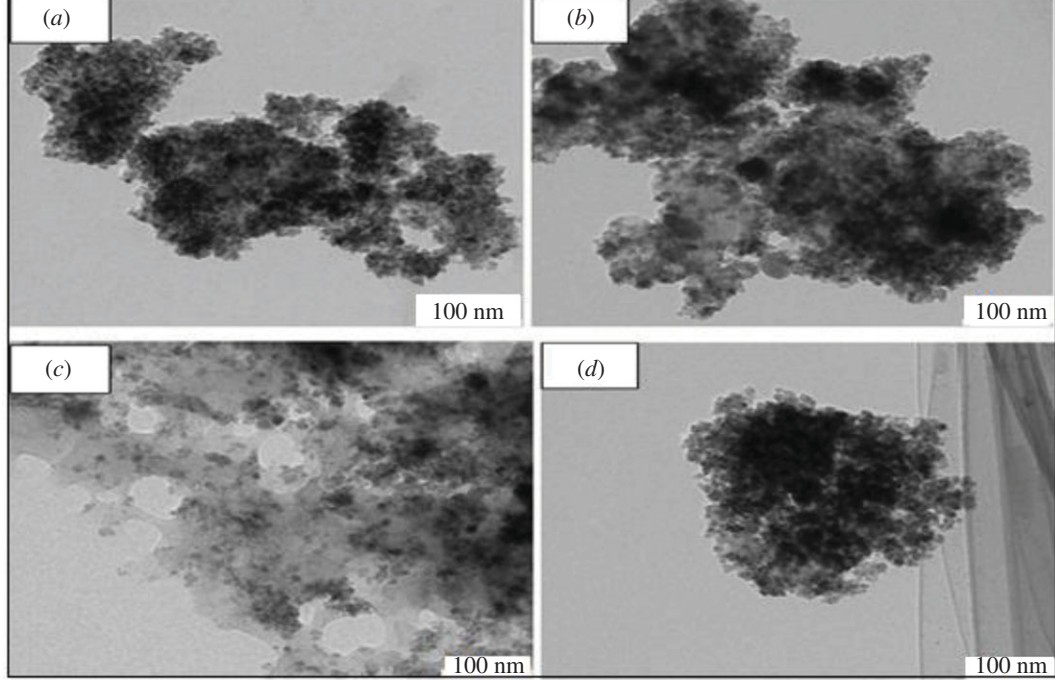

**Figure 6.** TEM images of (*a*) MMIP MAA–βCD, (*b*) MMIP MAA, (*c*) MNIP MAA–βCD and (*d*) MMIP MAA.

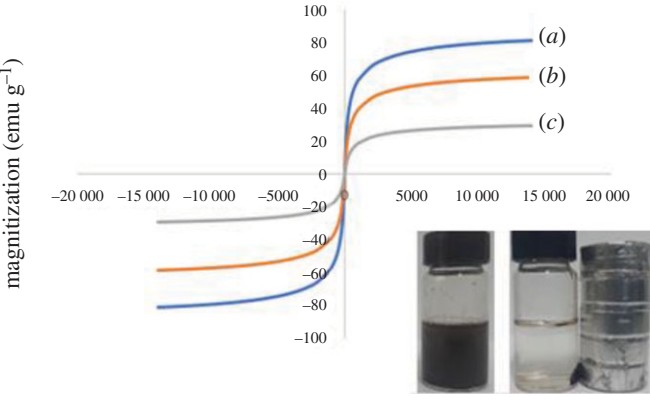

**Figure 7.** VSM magnetization curves of (*a*) bare Fe₃O₄, (*b*) MMIP MAA and (*c*) MMIP MAA–βCD.

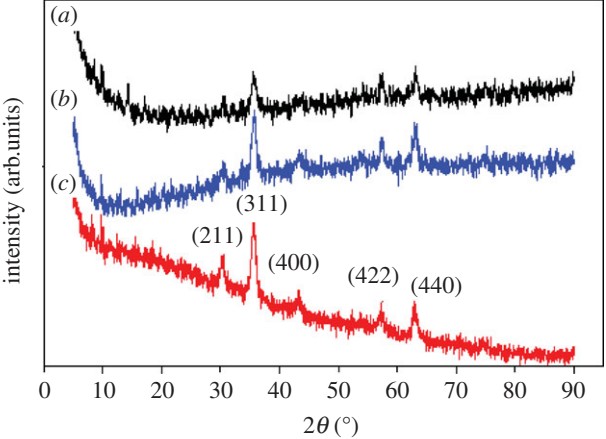

**Figure 8.** XRD patterns of (*a*) MMIP MAA–βCD, (*b*) MMIP MAA and (*c*) bare Fe₃O₄.

committee on powder diffraction standard (19-0629) [27]. The solid samples showed the angles corresponding to the peaks at $2\theta = 30.46$, 35.73, 43.51, 57.30 and 63.04, which were represented as (220), (311), (400), (422) and (440) cubic spinal planes of Fe₃O₄ particles. The intensity of the MMIP MAA–βCD peaks was seen to decrease owing to the amorphous β-CD layer. However, the size of the crystals was determined from the most intense peak (311) of the XRD diffraction pattern (figure 8) using the Scherrer equation described in equation (4.1):

$$D = \frac{K\lambda}{\beta \cos \theta} \qquad (4.1)$$

where $D$ is the diameter of the crystallite, $K$ represents the dimensionless shape factor (0.94), $\lambda$ is the X-ray wavelength ($\lambda$ Cu = 1.54178 Å), $\beta$ is the full width at half maximum (2.15) and $\theta$ denotes Braggs angle of reflection (35.73) from the most intense peak at $2\theta$. Substituting the values into the Scherrer equation obtained a crystallite size of 4.06 nm, which was consistent with the result obtained from TEM confirming the mesoporous nature of the adsorbent.

### 4.1.7. Inclusion complex of methacrylic acid-β-cyclodextrin-bisphenol A

The structure of the inclusion complex of MAA–βCD and BPA was investigated using ¹H NMR spectroscopy. The formation of the MAA–βCD/BPA inclusion complexes can be proved from the changes of chemical shifts in ¹H NMR spectra because the physical or chemical environment is affected by the hydrogen of BPA or the MAA–βCD cavity if the inclusion occurs. The ¹H NMR spectra of MAA–βCD and MAA–βCD–BPA in DMSO-d6 are depicted in figures 9 and 10. The ¹H chemical shift values of BPA before and after forming the inclusion complex are tabulated in the electronic supplementary material, table SD2. The result showed that the inclusion of BPA into the

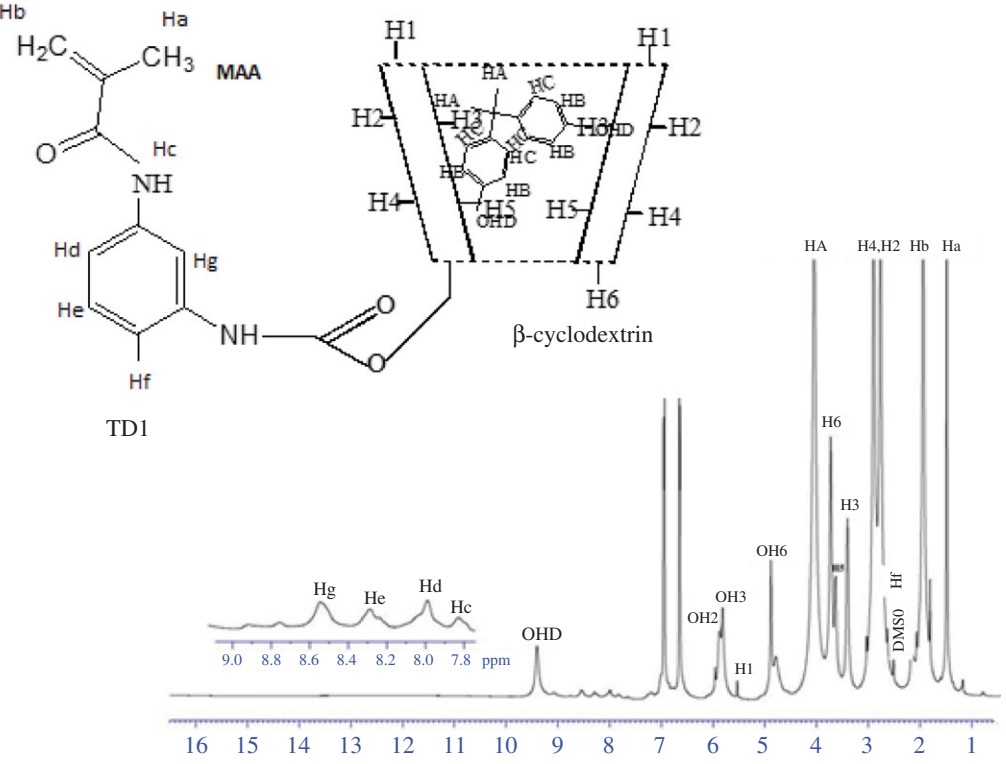

**Figure 9.** NMR spectrum of MAA–βCD monomer.

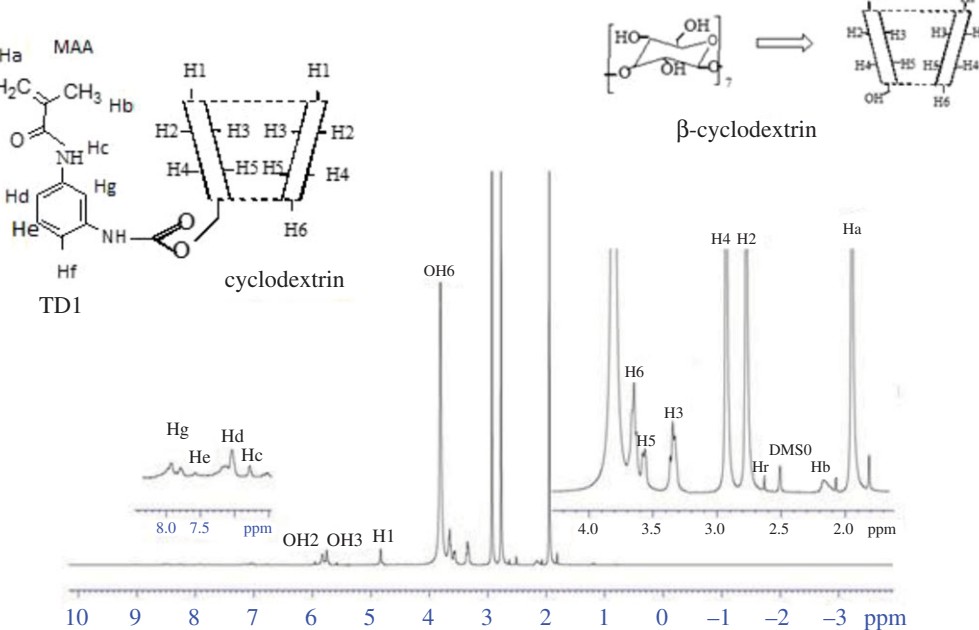

**Figure 10.** NMR spectrum of MAA–βCD–BPA inclusion complex.

cavity of βCD of the MAA–βCD monomer occurred owing to the weak but significant changes observed in the chemical shift for H5 (−0.261 ppm) and H3 (−0.052 ppm) protons, which are located inside the cavity of βCD. Conversely, H1, H2, H4 and H6 protons that were located at the exterior rim did not show much difference in chemical shift, thus confirming the formation of the inclusion complex between the MAA–βCD monomer and BPA. The alphatic protons designated as HA shifted upfield from 4.045 to 3.756 ppm and become more shielded. HB shifted downfield and are more deshielded owing to the presence of an electronegative oxygen atom. Also, HC shifted more downfield owing to the electronegative environment of the MAA–βCD cavity.

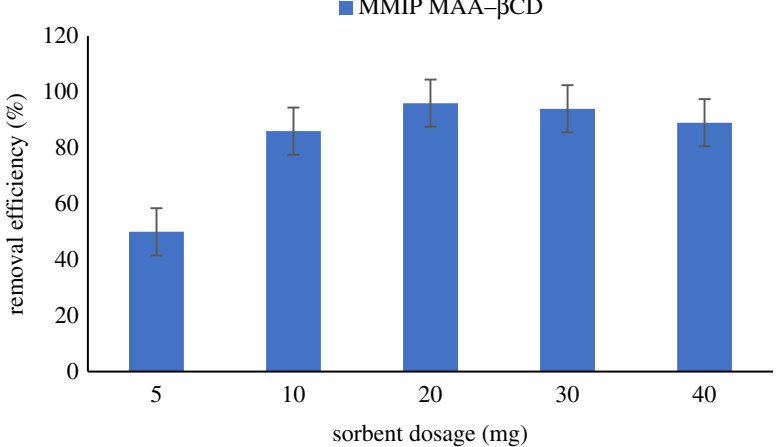

**Figure 11.** Effect of sorbent dosage on the removal of BPA from aqueous samples. Condition: 298 K, 10 ml of 10 mg l$^{-1}$ of analytes solution, 20 mg sorbent dose and 60 min shaking time at 250 r.p.m.

## 4.2. Optimization of adsorption parameters

### 4.2.1. Effect of adsorbent dose

The optimum dose of adsorbent to be used for adsorption can be maximized by adjusting the adsorbent dose when other parameters remain constant [22]. The effect of the sorbent dosage was studied at room temperature in this study to determine the maximum amount of sorbent to be used. The sorbent quantity used varied between 5 and 40 mg, while keeping other variables constant such as 60 min, 10 mg l$^{-1}$ and 250 r.p.m. for time, initial concentration and agitation speed, respectively. As shown in figure 11, the increase in the adsorbent's quantity allows the removal performance to increase. This was because active adsorption sites were available [28]. Removal performance increased steadily from 5 to 20 mg. Afterwards, there was a decrease in the removal performance because the vacant adsorption site has been completely occupied. Therefore, the quantity of adsorbent used in this study was set at 20 mg.

### 4.2.2. Effect of pH

The pH of the adsorption medium is the most critical parameter influencing the adsorption capacity [29]. This parameter may have an effect on both the electrical surface charge of the adsorbent and the dissociation constant ($pK_a$) of the adsorbate [30,31]. In this analysis, a pH of 2–11 has been selected. The required pH was adjusted using 0.01 M HCl and 0.01 M NaOH solutions. Adsorbent dose of 20 mg, adsorbate concentration of 10 mg l$^{-1}$ and agitation time of 60 min were used. Based on the findings presented in figure 12, the removal efficiency of the adsorbents increases steadily until its maximal value is reached at pH 8. The $pK_a$ value of BPA was 9.6–10.2. Generally, when the pH was lower than the $pK_a$, the BPA molecules would exist mainly in a molecular form, which is more hydrophobic than the ionic form. However, when the pH is more than $pK_a$, the dissociation form becomes more dominating, and as a result, the BPA molecules would exist mainly as negative bisphenolate anions, which eventually leads to the disappearance of the hydrogen bonds [35,36]. It is well known that β-CD can form inclusion complexes with hydrophobic compounds. It was established that the hydrophilic guest molecules (protonated or deprotonated) were unfavourable to form inclusion complexes with β-CD. Thus, only neutral forms of BPA were favourable to form inclusion complexes [37]. These findings indicated that BPA was trapped by the adsorbents through the hydrophobic interaction, inclusion complex formation and hydrogen-bonding concurrently between the imprinted sites and BPA. Therefore, pH 8 was adopted for further study.

### 4.2.3. Effect of contact time

Contact time is one of the crucial factors in the batch adsorption process [38]. The effect of contact time on the removal of BPA from aqueous solutions was carried out by varying contact time ranging from 10 to

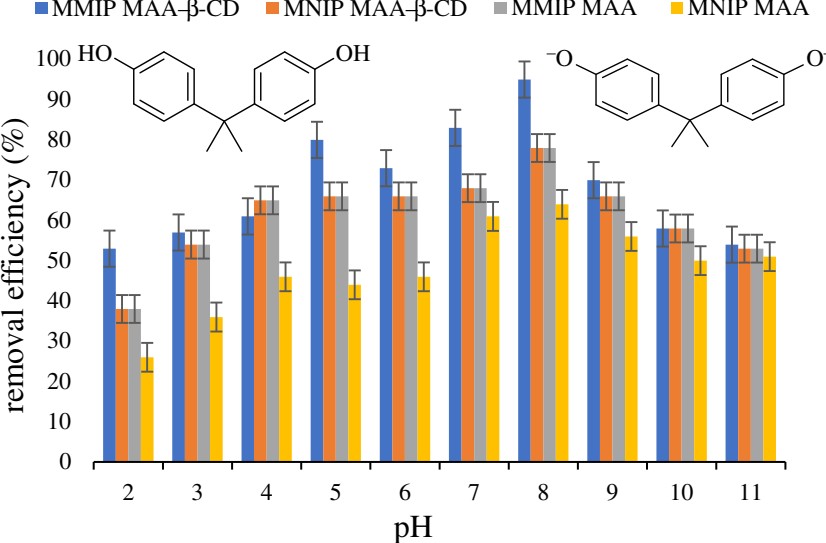

**Figure 12.** Effect of pH on BPA removal. Conditions: 298 K, 10 ml of 10 mg l$^{-1}$ of analyte solution, 20 mg sorbent dose and 60 min shaking time at 250 r.p.m.

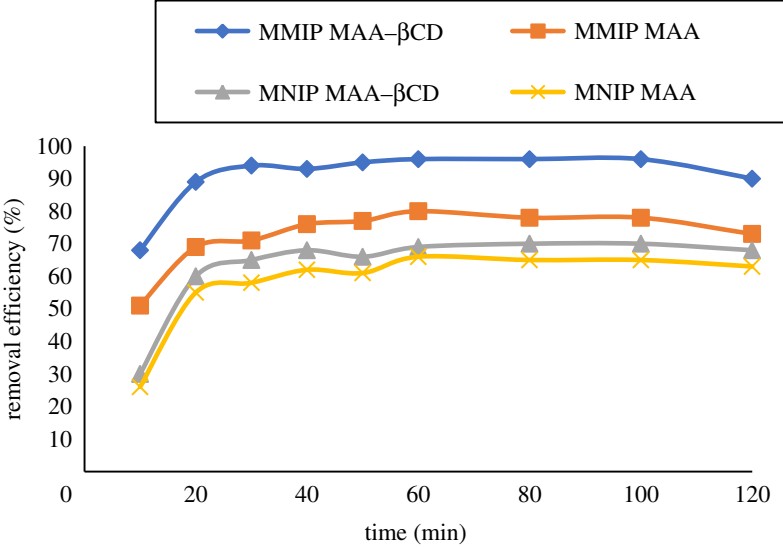

**Figure 13.** Effect of contact time on the removal of BPA onto MMIP MAA–βCD, MNIP MAA–βCD, MMIP MAA and MNIP MAA. Conditions: 298 K, 10 ml of 10 mg l$^{-1}$ of analyte solution at pH 8, 20 mg sorbent dose and 250 r.p.m. shaking speed.

120 min, while keeping the adsorbents amount, adsorbate concentration and pH constant. From the results obtained, the adsorption increased from 10 min and gradually increased until equilibrium was attained at 80 min, as shown in figure 13. The highest removal percentage was recorded at 60 min. The fast adsorption at the initial stage might be owing to the availability of abundant active sites and high driving force, both of which made BPA rapidly transfer to the adsorbent material. The decrease in removal efficiency could be owing to fewer and fewer driving forces. Therefore, 60 min was chosen as the optimized time in this study.

### 4.2.4. Effect of concentration and temperature

Temperature is an important parameter in adsorption studies. For any process, it is important to examine the impact of temperature on the efficiency of the adsorption process [39]. Experiments were conducted at various temperatures ranging from 298 to 328 K, while keeping all other parameters constant as shown in figure 14. The nature of the process can be understood depending on the way the process responds to

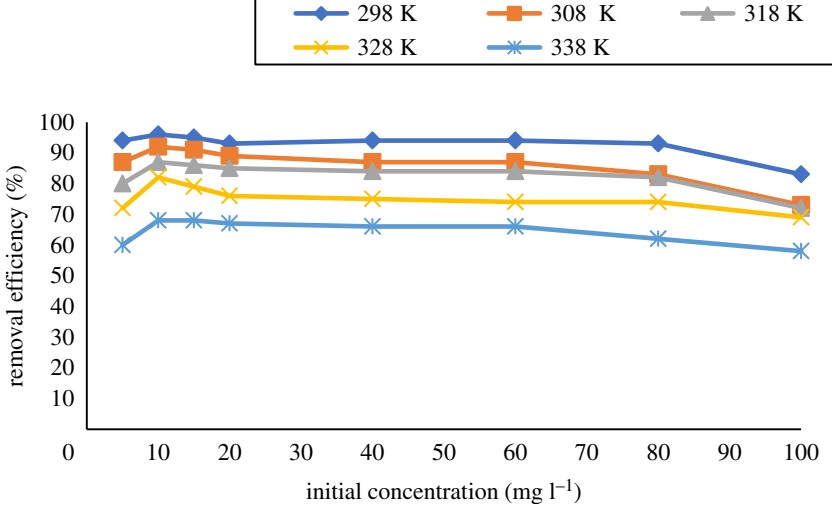

**Figure 14.** Effect of initial concentration and temperature on the removal of BPA. Conditions: 298 K, 10 ml of analyte solution at pH 8, 20 mg sorbent dose and 60 min shaking time at 250 r.p.m.

variation in applied temperature [39,40]. A process is considered as endothermic if there is an increase in temperature and exothermic if there is a decrease in the removal efficiency with an increase in temperature [41]. The experiment was carried out at initial analyte concentrations of 5–100 mg l$^{-1}$, sorbent of 20 mg, initial concentration of 10 mg l$^{-1}$, adsorption time of 60 min and stirring speed of 250 r.p.m. at pH 8 while varying the temperatures (298, 308, 318, 328 and 338 K).

## 4.3. Selectivity study

The selectivity study was conducted to further study the recognition properties of the polymers. Two phenol derivatives, 2,4-DCP and 2,4-DNP, were selected as potential interferences. Although 2,4-DCP and 2,4-DNP showed some affinity towards MMIP, the removal efficiency was lower than BPA. This indicates that the material had greater molecular recognition towards its template molecule [42]. The reason is because MMIP could recognize its template molecule owing to the presence of memory cavities of fixed size, shape, binding sites and specific binding interactions between the target molecule and sites. The selectivity of the synthesized MMIP MAA–βCD and MMIP MAA was compared (electronic supplementary material, figure SD3). The synthesized MMIP MAA–βCD has demonstrated a higher binding capacity towards BPA than MMIP MAA as presented in the electronic supplementary material, table SD3. This demonstrates that the presence of β-CD contributed to obtaining a higher binding capacity through the inclusion of complex formation and hydrophobic interaction. Hence, MMIP MAA–βCD was chosen as the best material owing to the higher imprinting effect towards BPA.

## 4.4. Reusability for adsorption study of bisphenol A onto magnetic molecularly imprinted polymer methacrylic acid-β-cyclodextrin

The reusability of imprinted polymers plays a crucial role in developing efficient, economical and sustainable applications. The ideal adsorbent must have a good adsorption capacity even after many cycles of reuse. This process not only lowers the cost but also helps to prevent waste generation. To investigate the ability of MMIP MAA–βCD to be reused numerous times, the adsorption was carried out in six cycles with the same adsorbents. It was discovered that the removal efficiency of the MMIP MAA–βCD remained high (above 90%) until the fifth cycle, indicating excellent stability, as shown in figure 15.

## 4.5. Adsorption kinetic study

It is well recognized that the kinetic models are often used to establish the nature of the adsorption system and to determine the stages that are influencing the process rate-limiting step, including chemical reaction and/or mass transfer [43].

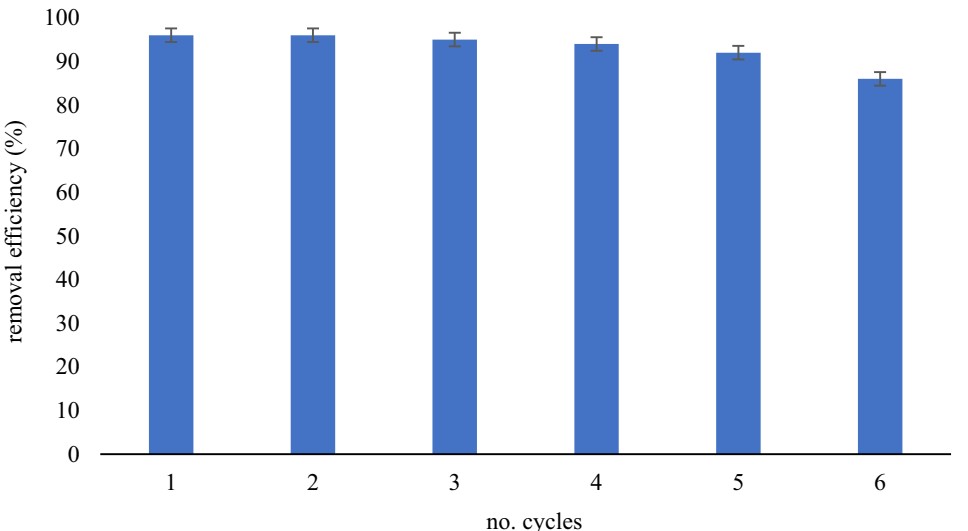

**Figure 15.** Reusability for adsorption of BPA onto MMIP MAA–βCD. Conditions: 298 K, 10 ml of 10 mg l$^{-1}$ of analyte solution at pH 8, 20 mg sorbent dose and 60 min shaking time at 250 r.p.m.

The result obtained showed that the pseudo second-order model better matched the experimental data for the adsorbents studied based on the high $R^2$-values > 0.9992, low $\Delta q$ % and relative error. It is well known that kinetic models are frequently used to determine adsorption mechanism and to evaluate the stages that affect the process rate-limiting step, including the chemical reaction and/or the mass transfer [40].

The results presented in table 1 revealed that the pseudo second-order model best suited the experimental results for the adsorbents tested based on high $R^2$-values > 0.9992, low $\Delta q$ % and relative error. Equilibrium adsorption capacity $q_e$ calculated complied with the $q_e$ obtained from the experimental data. This form of phenomenon has been documented in several studies on the adsorption of BPA [41–44]. The adsorption seemed to be driven by a chemical adsorption process through the sharing or exchange of electrons between adsorbent and adsorbate [32,44]. It may react with two forms of adsorption sites in the MMIP MAA–βCD and imprinted sites in the polymer network and may also form an inclusion complex with β-CD in the molecular recognition process [45]. Compared to MMIP MAA, the high initial sorption rate values ($h$) and low values of time required for the adsorption at $t_{1/2}$ proved that the MMIP MAA–βCD was a better adsorbent. This might occur owing to the existence of β-CD moiety in the internal surface of MMIP MAA–βCD, which gives a large pore diameter and lower pore volume of the polymer matrix, proving that the super pores improved the mass transport of the analyte. The adsorption system disagreed with the pseudo first-order model; the $q_e$ cal values were not in line with the experimental uptake $q_e$ exp.

The mechanism of the rate-limiting step in this sorption process was further investigated by fitting the experimental data into the intraparticle diffusion plot. According to this model, the plot of uptake ($q_t$) versus $t^{0.5}$ should be linear if the intraparticle diffusion was involved in the adsorption process. The value of C provides an idea of the thickness of the surface, and it is obtained from the slope and intercept [46]. The intercept, which was the thickness of the surface, gave information regarding the contribution of the surface adsorption in the rate-determining step. The larger the intercept, the greater its contribution. The value of C was highest for MMIP MAA–βCD (table 1), proving that it is the best adsorbent. In this study, the plot of $q_t$ versus $t^{0.5}$ did not pass through the origin, and this is an indication of some degree of boundary layer control and that the intraparticle diffusion is not the only rate limiting step, but that other kinetic models may control the rate of adsorption, all of which may be operating simultaneously [33]. The kinetic data fitted in the order of pseudo second-order > Elovich > intra-particle diffusion > external diffusion > pseudo first order.

## 4.6. Isotherm adsorption study

The adsorption isotherms offer an in-depth look at the relationship between the adsorbate and the adsorbent. In this study, adsorption data were tested with the linearized forms of five different

**Table 1.** Kinetic parameters for the adsorption of BPA onto MMIP MAA–βCD.

| kinetic models | parameters | materials | | | |
| --- | --- | --- | --- | --- | --- |
| | | MMIP MAA–βCD | MMIP MAA | MNIP MAA–βCD | MNIP MAA |
| | $q_e$ exp (mg g$^{-1}$) | 5.11 | 4.335 | 4.155 | 3.545 |
| pseudo first order | $q_e$ cal (mg g$^{-1}$) | 0.3831 | 0.5138 | 0.4085 | 0.9026 |
| | $K_1$ min$^{-1}$ | 0.00276 | 0.0269 | 0.00302 | 0.038 |
| | $R^2$ | 0.2611 | 0.2931 | 0.2763 | 0.4413 |
| | $\Delta q$ (%) | 37.7642 | 35.9861 | 36.8111 | 30.4303 |
| | relative error (%) | 95.5029 | 88.1476 | 88.3273 | 74.5388 |
| pseudo second order | $q_e$ cal (mg g$^{-1}$) | 5.1467 | 4.3802 | 4.191 | 3.6377 |
| | $K_2$ (g mg$^{-1}$ min) | 0.2828 | 0.1968 | 0.2533 | 0.1184 |
| | $h$ (mg g$^{-1}$ min) | 7.4906 | 3.775 | 4.4484 | 1.5672 |
| | $t^{1/2}$ (min) | 0.0549 | 0.0449 | 0.0604 | 0.0478 |
| | $R^2$ | 0.9999 | 0.9996 | 0.9999 | 0.9992 |
| | $\Delta q$ (%) | 0.2932 | 0.4257 | 0.3537 | 1.0675 |
| | relative error (%) | −0.7182 | −1.0427 | −0.8664 | −2.615 |
| Elovich | $q_e$ cal (mg g$^{-1}$) | 5.0892 | 4.2907 | 4.1418 | 3.4846 |
| | $\beta$ | 7.0274 | 5.5928 | 6.0277 | 3.9604 |
| | $\alpha$ | $5.6751 \times 10^{12}$ | $6.0512 \times 10^{9}$ | 1.3374 | $4.1454 \times 10^{12}$ |
| | $R^2$ | 0.981 | 0.9413 | 0.9437 | 0.9476 |
| | $\Delta q$ (%) | 0.1662 | 0.4172 | 0.1279 | 0.6956 |
| | relative error (%) | 0.8664 | 2.615 | 0.407 | 1.0219 |
| intra-particle diffusion | $C$ (mg g$^{-1}$) | 4.6546 | 3.7385 | 3.6356 | 2.6935 |
| | $K$ (mg g$^{-1}$ min) | 0.7182 | 1.0427 | 0.8664 | 2.615 |
| | $R^2$ | 0.9142 | 0.905 | 0.8756 | 0.948 |
| external diffusion | $K$ ext (1 min$^{-1}$) | 0.0127 | 0.005 | 0.004 | 0.0043 |
| | $C$ (mg g$^{-1}$) | 2.2683 | 1.3217 | 1.2467 | 0.801 |
| | $R^2$ | 0.8744 | 0.8313 | 0.7765 | 0.8885 |

isotherms models (Langmuir, Freundlich, Temkin, Halsey and Dubinin–Radushkevich (D–R)) to determine the model that best fits the equilibrium data.

The result obtained is shown in table 2. The Langmuir model, which is applied for monolayer adsorption on homogeneous systems, has been widely used to describe the sorption of solute from the water sample [47]. The coefficient of determination ($R^2$) obtained in this study was 0.4325–0.9229 at different temperatures studied. Therefore, it is not the best model to describe the adsorption of BPA. The Freundlich model best fitted the adsorption equilibrium data with $R^2 > 0.9563$, indicating MMIP MAA–βCD as a heterogeneous system. The $K_F$ values of 0.6841–8.0742 were obtained. The decrease in the $K_F$ value with an increase in the temperature proves the exothermic nature of the adsorption process. The value of $1/n$ is a measure of surface heterogeneity or adsorption intensity, and the surface becomes more heterogeneous when its value gets closer to zero [48]. In this study, the $n$ values above unity and $1/n$ values of less than unity at lower temperatures showed that the adsorbent was favourable, with a relatively strong bond formed between the adsorbate and the adsorbent [46,49]. Based on the coefficient of determination ($R^2$, 0.7703–0.9660), the Temkin model was evaluated for the removal of BPA onto MMIP MAA–βCD. According to the acquired results, this model does not fit well into the equilibrium data compared to the other models evaluated in this study. The constant $K_T$ (l mg$^{-1}$) and Temkin constant $b_T$ values were found to decrease when the

**Table 2.** Isotherm constants of five isotherm models for the adsorption of BPA onto MMIP MAA–βCD.

| MMIP MAA–βCD | | | | | | |
|---|---|---|---|---|---|---|
| | | temperature (K) | | | | |
| isotherm models | parameters | 298 | 308 | 318 | 328 | 338 |
| Langmuir | $q_m$ (mg g$^{-1}$) | 66.2252 | 75.7576 | 103.0928 | 76.9231 | 60.9756 |
| | $b$(l mg$^{-1}$) | 0.1472 | 0.0697 | 0.0269 | 0.02051 | 0.0150 |
| | $R^2$ | 0.9229 | 0.8263 | 0.5906 | 0.4718 | 0.4325 |
| | $R_L$ | 0.1017 | 0.1930 | 0.3826 | 0.4484 | 0.5263 |
| Freundlich | $K_F$ (mg g$^{-1}$) (l mg$^{-1}$)1/n | 8.0742 | 5.1928 | 2.9000 | 0.8493 | 0.6841 |
| | $n$ | 1.3910 | 1.3198 | 1.1562 | 0.8493 | 0.7852 |
| | $1/n$ | 0.7143 | 0.7577 | 0.8649 | 1.1774 | 1.2735 |
| | $R^2$ | 0.9917 | 0.9851 | 0.9686 | 0.9646 | 0.9563 |
| Temkin | $K_T$ (l mg$^{-1}$) | 3.2996 | 1.8528 | 0.9268 | 0.5149 | 0.2782 |
| | $b_T$(kJ mol$^{-1}$) | 239.77 | 247.32 | 236.42 | 123.89 | 112.55 |
| | $R^2$ | 0.9660 | 0.8835 | 0.9312 | 0.7703 | 0.8401 |
| Dubinin–Radushkevich | $q_m$ (mg g$^{-1}$) | 36.7927 | 21.6282 | 22.1447 | 21.6499 | 35.0894 |
| | $\beta$ (l mg$^{-1}$) | 0.6938 | 0.2696 | 0.8306 | 0.9516 | 5.7505 |
| | $R^2$ | 0.9481 | 0.7638 | 0.8339 | 0.5156 | 0.8064 |
| | $E$ | 2.4011 | 1.3618 | 0.7759 | 0.7249 | 0.2949 |
| Halsey | $n$ | 1.3999 | 1.3198 | 1.1562 | 0.8493 | 0.7852 |
| | $K_H$ | 0.0537 | 0.1137 | 0.2920 | 0.7088 | 0.7353 |
| | $R^2$ | 0.9917 | 0.9851 | 0.9686 | 0.9646 | 0.9563 |

temperature increased, which could evaluate the exothermic nature of the adsorption system. Like the Freundlich isotherm model, the Halsey model is suitable for multilayer adsorption as well as the heterogeneous surfaces in which the adsorption heat is non-uniformly distributed [50]. In this study, the Halsey model fits well with the equilibrium data ($R^2$, 0.9563–0.9917), confirming that MMIP MAA–βCD is a heterogeneous system characterized by the presence of different types of binding sites with different binding energies [51]. Conversely, the D–R isotherm model is related to the free energy ($E$) of adsorption per molecule of adsorbate when transferred to the surface of the solid from infinity in the solution. Thus, it allows us to predict the type of adsorption [52]. The D–R model reflected a poor fit to the experimental equilibrium data with $R^2$ values ranging from 0.5156 to 0.94810 compared to other models in this study. However, the $E$ values for the adsorption of BPA were low (0.2949–2.4011; table 2) for the spectrum of temperatures studied. Whenever $E$ is between 8 and 16 kJ mol$^{-1}$, the adsorption has been documented to have been chemical or ionic exchange-like, and when $E$ is between 1 and 8 kJ mol$^{-1}$, it implies physical adsorption [53]. Thus, the physisorption mechanism for adsorption of BPA prevailed in this study.

## 4.7. Thermodynamic study

Thermodynamic parameters calculated from the slope and intercept of the Van Hoff plot (electronic supplementary material, figure SD3) at different temperatures (298, 308, 318, 328 and 338 K) are presented in table 3. The negative standard free energy change ($\Delta G°$) indicates the feasibility and spontaneity of the adsorption process. However, the value of $\Delta G°$ becomes more negative with the reduction in temperature, signifying that lower temperature facilitates the adsorption of BPA onto MMIP MAA–βCD. The value of $\Delta H°$ also provides information about the nature of adsorption, which can either be physical or chemical [54]. The negative $\Delta H°$ (−18.9144) value suggested the exothermic nature of the adsorption process, which was also backed by the decline in the removal efficiency of

**Table 3.** Calculated values of thermodynamic parameters for the removal of BPA.

| $T$ (K) | Gibbs energy, $\Delta G°$ kJ mol$^{-1}$ | enthalpy, $\Delta H°$ J mol$^{-1}$ | entropy, $\Delta S°$ J K$^{-1}$ mol |
|---|---|---|---|
| 298 | −3751.9 | | |
| 308 | −2899.2 | −18.9144 | −51.723 |
| 318 | −1790.4 | | |
| 328 | −2702.9 | | |
| 338 | −1759.7 | | |

**Table 4.** Analytical performance data for adsorption study using UV−vis spectrophotometry.

| regression equation | $R^2$ | spiked concentration (mg l$^{-1}$) | intra-day ($n = 5$) | | inter-day ($n = 6$) | |
|---|---|---|---|---|---|---|
| | | | removal (%) | RSD (%) | removal (%) | RSD (%) |
| $Y = 0.0156x +$ | 0.9997 | 1 | 91.86 | 1.69 | 91.56 | 1.62 |
| 0.040 | | 10 | 95.57 | 0.60 | 95.63 | 0.58 |
| | | 60 | 92.29 | 1.03 | 93.13 | 1.32 |

**Table 5.** Removal efficiency (%) and RSD ($n = 3$) of real sample for the removal of selected phenols by MMIP MAA−βCD.

| samples | spiked volume (mg l$^{-1}$) | removal (%) | RSD (%) |
|---|---|---|---|
| plastics industry (A) | 1 | 93.09 | 0.62 |
| | 10 | 95.73 | 1.19 |
| | 60 | 94.81 | 1.84 |
| wooden industry | 1 | 92.51 | 0.49 |
| | 10 | 95.91 | 0.27 |
| | 60 | 94.34 | 0.53 |
| tap water | 1 | 92.36 | 0.28 |
| | 10 | 96.10 | 0.19 |
| | 60 | 94.49 | 0.26 |
| plastics industry (B) | 1 | 92.76 | 4.79 |
| | 10 | 96.11 | 3.40 |
| | 60 | 93.77 | 1.43 |
| rubber industry | 1 | 92.63 | 3.80 |
| | 10 | 96.28 | 2.52 |
| | 60 | 94.72 | 2.80 |

BPA and the decrease in $K_F$ value of the equilibrium isotherm data with an increase in the temperature. The result acquired in this study indicated that the adsorption mechanism was in the physisorption process. Standard entropy change ($\Delta S°$) was found to be negative in this study (−51.723 J K$^{-1}$ mol), reflecting decreased randomness at the solid–liquid interface during the adsorption of BPA onto the adsorbent [53]. Similar observations have been reported [55,56].

## 4.8. Method validation and real sample analysis

To evaluate the applicability of the developed method for the removal of phenolic compounds, five water samples (plastic industry A; wooden industry; plastic industry B; rubber industry from Prai, Malaysia,

industrial area; and a tap water sample from AMDI Universiti Sains Malaysia) were tested under the optimum MMIP MAA–βCD conditions. The method was validated by linearity, precision and removal analysis. The linear curve was obtained by spiking analyte solution in real water samples at a linear range of 1–60 mg l$^{-1}$ (representing low, medium and high concentrations). The $R^2$ of 0.9997 BPA was obtained through the linear curve. Repeatability studies were conducted for intra-day and inter-day. Intra-day precision was obtained from five consecutive replicates within the same day and expressed as RSD using three different vials ($n = 3$), and inter-day five replicates were analysed consecutively for 5 days and skipping 1 day to complete the removal on the sixth day ($n = 6$). The RSD was achieved in the range of 0.60–1.69% and 0.58–1.62% for intra-day and inter-day, respectively, as presented in table 4. The removal study was carried out in three replicates ($n = 3$) applied to three different concentrations of spiked samples. This is to verify the accuracy of the proposed method, and satisfactory removal was obtained (table 5). The result indicated that the method was successfully validated, and it is suitable for routine removal of phenolic compounds from aqueous media.

## 5. Conclusion

The use of cyclodextrin-based materials as adsorbents for the removal of different contaminants, which is the cutting-edge research, has caught the attention of many researchers worldwide. In this study, MMIP MAA–βCD, MNIP MAA–βCD, MMIP MAA and MNIP MAA were successfully synthesized for the removal of BPA in aqueous media by incorporating the benefit of molecular recognition and rapid magnetic response. Several variables influencing the adsorption efficiency of BPA have been analysed in depth, with an optimum adsorption time of 60 min at pH 8, 10 mg l$^{-1}$ analyte concentration, 20 mg adsorbent dose and 250 r.p.m. stirring speed. The pseudo second-order model provided the best fit for the kinetic results. The Freundlich and Halsey model best fitted the adsorption equilibrium data with ($R^2 > 0.9563$), indicating MMIP MAA–βCD is a heterogeneous system. Thermodynamic experiments have demonstrated that the adsorption system is thermodynamically feasible, exothermic and spontaneous. The result also revealed that the adsorption was mostly based on hydrogen bonding, interaction between the imprinted sites and the hydrophobic cavity of BPA. The high removal performance of the spiked real samples and the adsorption characteristics of the MMIP MAA–βCD, which is the best adsorbent, have shown that it is promising for routine BPA removal in aqueous samples.

Data accessibility. The datasets supporting this article have been uploaded as part of the electronic supplementary material. Data also available on Dryad: https://doi.org/10.5061/dryad.8pk0p2nmc [57].

Authors' contributions. S.M. carried out the adsorption study work, participated in data analysis and drafted the manuscript. F.B.M.S. and F.S.M. collected field data and revised the manuscript. M.R. helped in the characterization part and critically revised the manuscript. S.A. and N.N.M.Z. conceived of the study, designed the study, coordinated the study, critically revised the manuscript and helped draft the manuscript.

Competing interests. We declare we have no competing interests.

Funding. The authors would like to take this opportunity to express their gratitude to Fundamental Research Scheme, Ministry of Higher Education (MOHE), Malaysia (FRGS, 203.CIPPT.6711661) for assistance and financial support.

Acknowledgements. The authors also acknowledge the Advanced Medical and Dental Institute and School of Chemical Science, Universiti Sains Malaysia for the facilities provided.

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
