## [Peer Review File · Royal Society Open Science]

Review History

RSOS-201604.R0 (Original submission)

Review form: Reviewer 1

Is the manuscript scientifically sound in its present form?

Yes

Are the interpretations and conclusions justified by the results?

Yes

Is the language acceptable?

Yes

Do you have any ethical concerns with this paper?

No

Have you any concerns about statistical analyses in this paper?

No

Recommendation?

Major revision is needed (please make suggestions in comments)

Comments to the Author(s)

1. The characterization of the polymer can be further analyzed by adding nuclear magnetic resonance and mass spectrometry.
2. Please explain that the extraction efficiency of MMIP MAA- CD decreases at pH= 4.
3. The number of reusability experiments can be increased appropriately.

Review form: Reviewer 2**Is the manuscript scientifically sound in its present form?**

No

Are the interpretations and conclusions justified by the results?

Yes

Is the language acceptable?

Yes

Do you have any ethical concerns with this paper?

No

Have you any concerns about statistical analyses in this paper?

No

Recommendation?

Major revision is needed (please make suggestions in comments)

Comments to the Author(s)

This is quite well written paper in which selective molecularly imprinted magnetic polymeric sorbent (MMIP) for bisphenol A (BPA) removal from water samples is reported. The MMIP was synthesized by bulk polymerization using β -cyclodextrin hybridized with methacrylic acid (MAA- β -CD) and methacrylic acid (MMA) as monomers, trimethylpropane trimethacrylate was used as crosslinker and benzoyl peroxide as reaction initiator. The obtained MMIP with β -CD (MAA- β -CD) was compared with MMIP MMA. The obtained sorbents were exactly characterized using different methods (FT-IR, SEM, TEM, VSM, BET and XRD). Also the kinetic of the sorption process and selectivity were determined and the obtained data show that the pseudo second order kinetic and Freundlich model well describe the process. The method was validated and used for real sample analysis. The Authors concluded that in optimal conditions MMIP MAA- β -CD was the best adsorbent for BPA and that the adsorption was mostly based on hydrogen bonding, interaction between the imprinted sites and hydrophobic cavity of BPA. Additionally the reusability of obtained imprinted polymers were determined and show good stability after five cycles.

My substantial comments are as follows:

- The introduction did not present the "state of the art", especially in the field of molecularly imprinted polymers synthesis and applications. The Authors presented many different papers unrelated to the topic of work writing that they concern MIPs for BPA (eg. Ref. 22 dealing with bilirubin or 28 with aspirin sorption) therefore some of them can be removed.

From the other side, some important papers from this range presented in the literature were omitted eg.:

Alnaimat, A.S., *J. Microchem. J.*, 2019, 147, 598

Tian M., *Analytical Methods*, 2019, 11, 4761

A. Poliwoda, et al., *Ecol. Chem. Eng. S*, 2016, 23, 651

or reviews

Zhou, T., *TRAC*, 2019, 114, 11

Turiel, E. Martin-Esteban, A., *TRAC*, 2019, 114, 574

Therefore the introduction the aim and particularly novelty of presented work should be pointed clearly.

Decision letter (RSOS-201604.R0)

Dear Dr MOHAMAD ZAIN:

Title: Removal of BPA from aqueous media using a highly selective adsorbent of hybridization cyclodextrin with magnetic molecularly imprinted polymer

Manuscript ID: RSOS-201604

The editor assigned to your manuscript has now received comments from reviewers. We would like you to revise your paper in accordance with the referee and Subject Editor suggestions which can be found below (not including confidential reports to the Editor). Please note this decision does not guarantee eventual acceptance.

Please submit your revised paper before 25-Dec-2020. Please note that the revision deadline will expire at 00.00am on this date. If we do not hear from you within this time then it will be assumed that the paper has been withdrawn. In exceptional circumstances, extensions may be possible if agreed with the Editorial Office in advance. We do not allow multiple rounds of revision so we urge you to make every effort to fully address all of the comments at this stage. If deemed necessary by the Editors, your manuscript will be sent back to one or more of the original reviewers for assessment. If the original reviewers are not available we may invite new reviewers.

On behalf of the Subject Editor Professor Anthony Stace and the Associate Editor Dr Nadia Martinez Villegas.

RSC Associate Editor:

Comments to the Author:

The research presented in this draft is original and of interest to RSOS audience, however additional characterization and experiments were suggested to carry out. Additionally, the introduction section should be improved.

Few additional specific comments include:

The parameters studied in this work should be mentioned in the abstract.

Page 4, line 11: 126 billion is higher than 2.9 billion. Please correct this.

Section 2.2. What was the database used to analyse XRD data? Please add this.

Page 6, lines 5 and 6: the stoichiometric ratio is repeated.

Page 7, lines 5 to 12: this paragraph is repeated.

Section 2.5. Please add some lines to explain the QA/QC analyses.

Page 7, line 35: Add “ (“ before 3.

Page 9: where should Figure 2 go?

Page 10, lines 34 and 41: Please correct the question mark

Page 21, Figure 1: What is the meaning of the blue color?

Page 23, Figure 5: What is the meaning of the red circle?

RSC Subject Editor:

Comments to the Author:

(There are no comments.)

Reviewers' Comments to Author:

Reviewer: 1

Comments to the Author(s)

1.The characterization of the polymer can be further analyzed by adding nuclear magnetic resonance and mass spectrometry.

2.Please explain that the extraction efficiency of MMIP MAA- CD decreases at pH= 4.

3.The number of reusability experiments can be increased appropriately.

Reviewer: 2

Comments to the Author(s)

This is quite well written paper in which selective molecularly imprinted magnetic polymeric sorbent (MMIP) for bisphenol A (BPA) removal from water samples is reported. The MMIP was synthesized by bulk polymerization using β -cyclodextrin hybridized with methacrylic acid (MAA- β -CD) and methacrylic acid (MMA) as monomers, trimethylpropane trimethacrylate was used as crosslinker and benzoyl peroxide as reaction initiator. The obtained MMIP with β -CD (MAA- β -CD) was compared with MMIP MMA. The obtained sorbents were exactly characterized using different methods (FT-IR, SEM, TEM, VSM, BET and XRD). Also the kinetic of the sorption process and selectivity were determined and the obtained data show that the pseudo second order kinetic and Freundlich model well describe the process. The method was validated and used for real sample analysis. The Authors concluded that in optimal conditions MMIP MAA- β -CD was the best adsorbent for BPA and that the adsorption was mostly based on hydrogen bonding, interaction between the imprinted sites and hydrophobic cavity of BPA. Additionally the reusability of obtained imprinted polymers were determined and show good stability after five cycles.

My substantial comments are as follows:

- The introduction did not present the "state of the art", especially in the field of molecularly imprinted polymers synthesis and applications. The Authors presented many different papers unrelated to the topic of work writing that they concern MIPs for BPA (eg. Ref. 22 dealing with bilirubin or 28 with aspirin sorption) therefore some of them can be removed. From the other side, some important papers from this range presented in the literature were omitted eg.:

Alnaimat, A.S., *J. Microchem. J.*, 2019, 147, 598

Tian M., *Analytical Methods*, 2019, 11, 4761

A. Poliwoda, et al., *Ecol. Chem. Eng. S*, 2016, 23, 651

or reviews

Zhou, T., *TRAC*, 2019, 114, 11

Turiel, E. Martin-Esteban, A., *TRAC*, 2019, 114, 574

Therefore the introduction the aim and particularly novelty of presented work should be pointed clearly.

Author's Response to Decision Letter for (RSOS-201604.R0)

See Appendix A.

RSOS-201604.R1 (Revision)

Review form: Reviewer 1

Is the manuscript scientifically sound in its present form?

Yes

Are the interpretations and conclusions justified by the results?

Yes

Is the language acceptable?

Yes

Do you have any ethical concerns with this paper?

No

Have you any concerns about statistical analyses in this paper?

No

Recommendation?

Accept as is

Comments to the Author(s)

It is good!

Review form: Reviewer 2

Is the manuscript scientifically sound in its present form?

Yes

Are the interpretations and conclusions justified by the results?

Yes

Is the language acceptable?

Yes

Do you have any ethical concerns with this paper?

No

Have you any concerns about statistical analyses in this paper?

No

Recommendation?

Accept as is

Comments to the Author(s)

The manuscript was revised taking into account most of my corrections, suggestions and comments and the reviewer 1 comments.

Based on the above, I can recommend this version of paper for publication in presented form

Decision letter (RSOS-201604.R1)

Dear Dr MOHAMAD ZAIN:

Title: Removal of bisphenol A from aqueous media using a highly selective adsorbent of hybridization cyclodextrin with magnetic molecularly imprinted polymer
Manuscript ID: RSOS-201604.R1

It is a pleasure to accept your manuscript in its current form for publication in Royal Society Open Science. The chemistry content of Royal Society Open Science is published in collaboration with the Royal Society of Chemistry.

On behalf of the Subject Editor Professor Anthony Stace and the Associate Editor Dr Nadia Martinez Villegas.

RSC Associate Editor:
Comments to the Author:
Your revisions have fulfilled the requirements. Thank you for considering RSOS for publication.

RSC Subject Editor:
Comments to the Author:
(There are no comments.)

Reviewer(s)' Comments to Author:
Reviewer: 2
Comments to the Author(s)
The manuscript was revised taking into account most of my corrections, suggestions and comments and the reviewer 1 comments.

Based on the above, I can recommend this version of paper for publication in presented form

Reviewer: 1

Comments to the Author(s)

It is good!

Appendix A

Reviewers Comments and Authors Response

Manuscript ID: RSOS-201604

Title: Removal of BPA from aqueous media using a highly selective adsorbent of hybridization cyclodextrin with magnetic molecularly imprinted polymer

Authors: S. Mamman^{1,4}, F. B. M. Suah¹, M. Raaov³, F.S. Mehamod⁵, S. Asman⁶, N. N. M. Zain^{2*}

***Corresponding author:** N. N. M. Zain

The authors would like to thank the editor and the reviewers for their precious time and invaluable comments. We have carefully addressed all the comments. The corresponding changes and refinements made in the revised paper are summarized in our response below.

Editor

1. The introduction section should be improved.

Reply:

Thank you for the comment. We are agreed to the comment. Thus, we have revised the introduction section.

2. The parameters studied in this work should be mentioned in the abstract.

Reply:

Thank you for the comment. We are agreed to the comment. Thus, the studied parameters were included in the abstract.

3. Page 4, line 11: 126 billion is higher than 2.9 billion. Please correct this.

Reply:

Thank you for the comment. We are agreed to the comment. The sentence has been revised accordingly.

4. Section 2.2. What was the database used to analyse XRD data? Please add this.

Reply:

Thank you for the comment. The information has been added.

5. Page 6, lines 5 and 6: the stoichiometric ratio is repeated.

Reply:

Thank you for the comment. We are agreed to the comment. The sentence has been revised.

6. Page 7, lines 5 to 12: this paragraph is repeated.

Reply:

Thank you for the comment. We are agreed to the comment. The repeated paragraph has been deleted.

7. Section 2.5. Please add some lines to explain the QA/QC analyses.

Reply:

Thank you for the comment. We are agreed to the comment. The explanation of QA/QC has been added.

8. Page 7, line 35: Add “(“ before 3.

Reply:

Thank you for the comment. The sentence has been corrected.

9. Page 9: where should Figure 2 go?

Reply:

Thank you for the comment. The position of Figure 2 has been added in the manuscript.

10. Page 10, lines 34 and 41: Please correct the question mark

Reply:

Thank you for the comment. We have corrected the question mark.

11. Page 21, Figure 1: What is the meaning of the blue color?

Reply:

Thank you for the comment. The blue colour is referred to the surface of adsorbent that has been covered by Fe_3O_4 particles. The colour has been changed to grey colour.

12. Page 23, Figure 5: What is the meaning of the red circle?

Reply:

Thank you for the comment. The red cycle has been removed from the image.

Reviewer 1

1. The characterization of the polymer can be further analyzed by adding nuclear magnetic resonance and mass spectrometry.

Reply:

Thank you for the comment. We are agreed to the comment. The characterization of the polymer via NMR has been added.

2. Please explain that the extraction efficiency of MMIP MAA- β CD decreases at pH= 4.

Reply:

Thank you for the comment. The explanation of pH has been revised and added.

3. The number of reusability experiments can be increased appropriately.

Reply:

Thank you for the comment. We are agreed to the comment. The reusability was conducted up to sixth cycles, however the removal efficiency dropped to 86% showing the material was stable up to fifth cycles.

Reviewer 2

1. The introduction did not present the "state of the art", especially in the field of molecularly imprinted polymers synthesis and applications.

Reply:

Thank you for the comment. We are agreed to the comment. The introduction has been revised accordingly.

2. The Authors presented many different papers unrelated to the topic of work writing that they concern MIPs for BPA (eg. Ref. 22 dealing with bilirubin or 28 with aspirin sorption) therefore some of them can be removed. From the other side, some important papers from this range presented in the literature were omitted eg.:
Alnaimat, A.S., J. Microchem. J., 2019, 147, 598
Tian M., Analytical Methods, 2019, 11, 4761
A. Poliwoda, et al., Ecol. Chem. Eng. S, 2016, 23, 651
or reviews
Zhou, T., TRAC, 2019, 114, 11
Turiel, E. Martin-Esteban, A., TRAC, 2019, 114, 574

Reply:

Thank you for the comment. We are agreed to the comment. The unrelated references have been removed and replaced with the suggested references.

3. Therefore, the introduction the aim and particularly novelty of presented work should be pointed clearly.

Reply:

Thank you for the comment. We are agreed to the comment. The introduction of aim and particularly novelty of presented work has been mentioned in the manuscript.